# Novel Fuzzy PID-Type Iterative Learning Control for Quadrotor UAV

**DOI:** 10.3390/s19010024

**Published:** 2018-12-21

**Authors:** Jian Dong, Bin He

**Affiliations:** College of Electronics and Information Engineering, Tongji University, Shanghai 200000, China; abc1710330@tom.com

**Keywords:** iterative learning control, proportional-interactive-derivative (PID), fuzzy control, quadrotor unmanned aerial vehicle (UAV), trajectory tracking

## Abstract

Due to the under-actuated and strong coupling characteristics of quadrotor aircraft, traditional trajectory tracking methods have low control precision, and poor anti-interference ability. A novel fuzzy proportional-interactive-derivative (PID)-type iterative learning control (ILC) was designed for a quadrotor unmanned aerial vehicle (UAV). The control method combined PID-ILC control and fuzzy control, so it inherited the robustness to disturbances and system model uncertainties of the ILC control. A new control law based on the PID-ILC algorithm was introduced to solve the problem of chattering caused by an external disturbance in the ILC control alone. Fuzzy control was used to set the PID parameters of three learning gain matrices to restrain the influence of uncertain factors on the system and improve the control precision. The system stability with the new design was verified using Lyapunov stability theory. The Gazebo simulation showed that the proposed design method creates effective ILC controllers for quadrotor aircraft.

## 1. Introduction

The quadrotor [1,2,3], as a branch of unmanned aerial vehicles (UAVs), is highly favored in both military and civilian applications given its vertical take-off and landing ability, insensibility to varying environments, high mobility and stability, and easy operation modes. The quadrotor system is highly-coupled, under-actuated, and inherently non-linear, which challenges the system stability involving the microprocessor, the sensor, the mechanism, and the navigation and control algorithm. Over the past few years, much work has been completed on the modeling [4,5] and control [6,7,8] of the quadrotor UAV. The concept was introduced as early as 1907. Since then, the theoretical and experimental research results on the aspects of posture balance and perfect trajectory tracking have been extensively reported, such as adaptive control [9], fuzzy control [10], optimal control [11], and loop shaping theory [12]. Bouadi et al. [13] used a sliding mode control algorithm based on the reverse step to control the aircraft and derived the attitude angle from the higher-order nonholonomic constraints, but the change in the position loop was not used as feedback in real time. Shakev et al. [14] applied the linear feedback method to achieve the steady control of the quadrotor aircraft but did not consider aerodynamic interference. Courbon et al. [15] used novel navigation and positioning to control the quadrotor aircraft in position and attitude control.

The quadrotor UAV was chosen in this study as the research object. We attached importance to the UAV model and the control algorithm to improve the robustness and stability of the system. Considering uncertainty and random disturbance in the process of aircraft flight, we used the iterative learning control (ILC) method to improve system robustness. The ILC [16,17,18] method has a simple and clear form, and can also compensate for uncertainty, nonlinearity, coupling, modeling error, and other factors through the online learning process. The advantage of using ILC is that there is no need for accurate knowledge of the quadrotor aircraft or physical parameters of the system. However, there has been relatively little focus on the ILC of quadrotor aircraft. Angela et al. [19] applied ILC to the trajectory tracking of a quadrotor. Ma et al. [20] combined the ILC and PD algorithm for the attitude tracking control of a reference attitude trajectory. We think it will be extended to include the PID-ILC scheme for quadrotor systems in order to improve the tracking performance and vibration control. 

Our proposed PID algorithm, which differs from the common PID algorithm [21,22,23] in the literature, uses the integral and differential of the derivative of the current error to improve the tracking performance of the system. Juan et al. [24] proposed a nonlinear robust PID controller for attitude regulation of the quadrotor. Ahmet et al. [25] presented a fault-tolerant PID control scheme for nonlinear quadcopter system to guarantee the stability of attitude and path control. Meanwhile, fuzzy control was added to the proposed algorithm. Fuzzy control [26,27,28] is fast and highly stable and can adjust the gain of the control algorithm according to the needs of the ILC to improve the convergence speed and tracking the accuracy of the ILC. The proposed scheme combines the PID-ILC with fuzzy control, and fuzzy control optimizes the parameters of the ILC law to find the optimal gain so that the algorithm can learn faster, and the system can accurately converge to the desired path with fewer iterations.

The paper is structured as follows: Section 2 depicts the complete model of the quadrotor UAV. Section 3 presents the fuzzy PID-ILC algorithm used to control the UAV. Section 4 presents the convergence analysis of the proposed algorithm. In Section 5, the Gazebo simulation results are provided. Finally, our conclusion is presented in Section 6.

## 2. Model for The Quadrotor UAV

The quadrotor model is described in this section. $E=\left\{X_{e},Y_{e},Z_{e}\right\}$ is the inertial coordinate system, $B=\left\{X_{b},Y_{b},Z_{b}\right\}$ is the body coordinate system, $\Phi ={\left[\varphi ,\theta ,\psi \right]}^{T}$ represents the Euler angle, and the rotation matrix from the inertial frame to the body frame is:(1)$$R=\left[\begin{array}{ccc} \cos\psi \cos\theta & \cos\psi \sin\varphi \sin\theta +\cos\varphi \sin\psi & \sin\varphi \sin\psi -\cos\psi \cos\varphi \sin\theta \\ -\cos\theta \sin\psi & \cos\varphi \cos\psi -\sin\psi \sin\varphi \sin\theta & \sin\psi \cos\varphi \sin\theta +\sin\varphi \cos\psi \\ \sin\theta & -\cos\theta \sin\varphi & \cos\theta \cos\varphi \end{array}\right]$$

According to Newton’s law of motion and the Euler equation, the dynamic equation of the quadrotor can be expressed as
(2)$$F=m\ddot{P}$$
(3)M=dH/dt
where F is the external force on the quadrotor, m is the mass of the quadrotor, M is the rotational torque of the airframe, H is the angular momentum of the body under the inertial coordinate system, l represents the distance from the motor shaft to the center of the body, $J_{r}$ represents the inertia of the motor, $f_{i}\left(i=1,2,3,4\right)$ represents the lift provided by the i-th rotor, and b and d represent the lift and drag coefficients of the rotors, respectively. J represents the inertia matrix of the airframe, $K_{\mathrm{dm}}$ represents the coefficient of rotational resistance moment, and $K_{\mathrm{dt}}$ represents the coefficient of translational resistance. According to the structural characteristics of the quadrotor UAV, J, $K_{\mathrm{dm}}$, and $K_{\mathrm{dt}}$ can be expressed as diagonal arrays:(4)$$J=\left[\begin{array}{ccc} I_{x} & 0 & 0\\ 0 & I_{y} & 0\\ 0 & 0 & I_{z} \end{array}\right],K_{\mathrm{dm}}=\left[\begin{array}{ccc} K_{\mathrm{dm}x} & 0 & 0\\ 0 & K_{\mathrm{dm}y} & 0\\ 0 & 0 & K_{\mathrm{dm}z} \end{array}\right],K_{\mathrm{dt}}=\left[\begin{array}{ccc} K_{\mathrm{dt}x} & 0 & 0\\ 0 & K_{\mathrm{dt}y} & 0\\ 0 & 0 & K_{\mathrm{dt}z} \end{array}\right]$$

$P={\left[x,y,z\right]}^{T}$ represents the position of the quadrotor centroid in the inertial coordinate system and $\Omega ={\left[p,q,r\right]}^{T}$ represents the rotational angular velocity around three axes in the body coordinate system, expressed as
(5)$$\begin{array}{l} \ddot{x}=\frac{\left(\cos\varphi \cos\psi \sin\theta +\sin\varphi \sin\psi \right)}{m}\sum_{i=1}^{4}f_{i}-\frac{K_{\mathrm{dt}x}}{m}\dot{x}\\ \ddot{y}=\frac{\left(\cos\varphi \sin\psi \sin\theta -\sin\varphi \cos\psi \right)}{m}\sum_{i=1}^{4}f_{i}-\frac{K_{\mathrm{dt}y}}{m}\dot{y}\\ \ddot{z}=\frac{\cos\varphi \cos\theta }{m}\sum_{i=1}^{4}f_{i}-\frac{K_{\mathrm{dt}z}}{m}\dot{z}-g \end{array}$$
(6)$$\begin{array}{l} \dot{p}=\left[qr\left(I_{y}-I_{z}\right)+lb\left(\omega_{4}^{2}-\omega_{2}^{2}\right)-K_{\mathrm{dm}x}p+J_{r}q\left(\omega_{1}+\omega_{3}-\omega_{2}-\omega_{4}\right)\right]/I_{x}\\ \dot{q}=\left[pr\left(I_{z}-I_{x}\right)+lb\left(\omega_{3}^{2}-\omega_{1}^{2}\right)-K_{\mathrm{dm}y}q+J_{r}p\left(\omega_{2}+\omega_{4}-\omega_{1}-\omega_{3}\right)\right]/I_{y}\\ \dot{r}=\left[qp\left(I_{x}-I_{y}\right)+d\left(\omega_{1}^{2}+\omega_{3}^{2}-\omega_{4}^{2}-\omega_{2}^{2}\right)-K_{\mathrm{dm}z}r\right]/I_{z} \end{array}$$
where $f_{i}=b\omega_{i}^{2}$, $\omega =\omega_{1}-\omega_{2}+\omega_{3}-\omega_{4}$, and $\omega_{i}$ is the rotational angular velocity of the rotor of the i-th rotor. $I_{x},I_{y},I_{z}$ represent the axial inertial moment of the aircraft in the *x*, *y*, and *z* directions, respectively. Equations (5) and (6) describe the centroid translational motion and the body rotation motion of the quadrotor UAV, respectively. The following relationship exists between the Euler angular velocity and angular velocity in the body coordinate system:(7)$$\left[\begin{array}{c} \dot{\varphi }\\ \dot{\theta }\\ \dot{\psi } \end{array}\right]=\left[\begin{array}{ccc} 1 & \sin\varphi \tan\theta & \cos\varphi \tan\theta \\ 0 & \cos\varphi & -\sin\varphi \\ 0 & \sin\varphi /\cos\theta & \cos\varphi /\cos\theta \end{array}\right]\left[\begin{array}{c} p\\ q\\ r \end{array}\right]$$

When the quadrotor is hovering or flying at low speeds indoors, we define vectors as $U={\left[\begin{array}{cccc} u_{1} & u_{2} & u_{3} & u_{4} \end{array}\right]}^{T}$.$u_{1},u_{2},u_{3},u_{4}$ represent the lift torque, roll torque, pitch torque, and yaw torque of the aircraft, respectively, and are defined as:(8)$$\begin{array}{l} u_{1}=b\left(\omega_{1}^{2}+\omega_{2}^{2}+\omega_{3}^{2}+\omega_{4}^{2}\right)\\ u_{2}=b\left(\omega_{2}^{2}-\omega_{4}^{2}\right)\\ u_{3}=b\left(\omega_{1}^{2}-\omega_{3}^{2}\right)\\ u_{4}=d\left(\omega_{1}^{2}-\omega_{2}^{2}+\omega_{3}^{2}-\omega_{4}^{2}\right) \end{array}$$

The following simplified model of the quadrotor UAV can be obtained:(9)$$\left\{ \begin{array}{l} \ddot{x}=\frac{u_{1}}{m}\left(\sin\psi \sin\varphi +\cos\psi \sin\theta \cos\varphi \right)\\ \ddot{y}=\frac{u_{1}}{m}\left(-\cos\psi \sin\varphi +\sin\psi \sin\theta \cos\varphi \right)\\ \ddot{z}=\frac{u_{1}}{m}\cos\theta \cos\varphi +g\\ \ddot{\varphi }=\frac{I_{y}-I_{z}}{I_{x}}\dot{\theta }\dot{\psi }-\frac{J_{r}}{I_{x}}\dot{\theta }\omega +\frac{u_{2}}{I_{x}}\\ \ddot{\theta }=\frac{I_{z}-I_{x}}{I_{y}}\dot{\varphi }\dot{\psi }-\frac{J_{r}}{I_{y}}\dot{\varphi }\omega +\frac{u_{3}}{I_{y}}\\ \ddot{\psi }=\frac{I_{x}-I_{y}}{I_{z}}\dot{\varphi }\dot{\theta }+\frac{u_{4}}{I_{z}} \end{array}\right.$$
where x,y,z,φ,θ,ψ represent the longitudinal displacement, lateral displacement, height, roll angle, pitch angle, and yaw angle of the aircraft, respectively, and g is the gravitational acceleration. The physical parameters for the quadrotor are provided in Table 1.

The quadrotor aircraft relies on the four rotors to generate lift and torque, enabling lifting, yaw, roll, pitch, lateral, and transverse movements. Its four propeller crosses are driven by four direct current (DC) motors, and motion in space is achieved by changing the speed of the four DC motors. The structure diagram of the quadrotor aircraft is shown in Figure 1.

To facilitate the formula derivation, we simplified the aircraft model. The state variable is $x=(\begin{array}{cccccccccccc} x & \dot{x} & y & \dot{y} & z & \dot{z} & \psi & \dot{\psi } & \theta & \dot{\theta } & \varphi & \dot{\varphi } \end{array})$, and the virtual input is $U=\left(\begin{array}{cccccc} U_{1} & U_{2} & U_{3} & U_{4} & U_{5} & U_{6} \end{array}\right)$. The mathematical model of the quadrotor was rewritten into an equation of the state format:(10)$$\left\{ \begin{array}{l} {\dot{x}}_{1}=x_{2}\\ {\dot{x}}_{2}=\frac{U_{5}U_{1}}{m}\\ {\dot{x}}_{3}=x_{4}\\ {\dot{x}}_{4}=\frac{U_{6}U_{1}}{m}\\ {\dot{x}}_{5}=x_{6}\\ {\dot{x}}_{6}=\frac{\cos x_{11}\cos x_{9}U_{1}}{m}-g\\ {\dot{x}}_{7}=x_{8}\\ {\dot{x}}_{8}=\frac{x_{10}x_{12}\left(I_{x}-I_{y}\right)}{I_{z}}+\frac{U_{4}}{I_{z}}\\ {\dot{x}}_{9}=x_{10}\\ {\dot{x}}_{10}=\frac{x_{8}x_{12}\left(I_{z}-I_{x}\right)}{I_{y}}+\frac{J_{r}x_{12}\Omega }{I_{y}}+\frac{U_{3}}{I_{y}}\\ {\dot{x}}_{11}=x_{12}\\ {\dot{x}}_{12}=\frac{x_{8}x_{10}\left(I_{y}-I_{z}\right)}{I_{x}}+\frac{J_{r}x_{10}\Omega }{I_{x}}+\frac{U_{2}}{I_{x}} \end{array}\right.$$
where $U_{5}=\left(\cos\phi \sin\theta \cos\psi +\sin\phi \sin\psi \right)$ and $U_{6}=\left(\cos\phi \sin\theta \cos\psi -\sin\phi \cos\psi \right)$. We divided the whole system into six relatively independent channels: height control, horizontal *X*-axis control, horizontal *Y*-axis control, roll control, pitch control, and yaw control.

The mathematical model of the height channel can be obtained by Equation (10):(11)$$\left\{ \begin{array}{l} {\dot{x}}_{5}=x_{6}\\ {\dot{x}}_{6}=\frac{\cos x_{11}\cos x_{9}U_{1}}{m}-g \end{array}\right.$$

The desired height is $z_{d}=x_{5d}$ and the tracking error is $e_{5}=x_{5d}-x_{5}$. We defined the Lyapunov function as $V_{5}=e_{5}^{2}/2$, and the derivative of the Lyapunov function is:(12)$${\dot{V}}_{5}=e_{5}{\dot{e}}_{5}=e_{5}\left({\dot{x}}_{5d}-x_{6}\right)$$

Set $x_{6}=\alpha_{5}+e_{6}$, $\alpha_{5}={\dot{x}}_{5d}+k_{5}e_{5}$, $k_{5}>0$ is parameter of control system and $\alpha_{5}$ is the virtual control input. Equation (12) can be simplified as:(13)$${\dot{V}}_{5}=-e_{5}e_{6}-k_{5}e_{5}^{2}$$

We defined the new Lyapunov function as $V_{6}=V_{5}+e_{6}^{2}/2$, and the derivative of this formula can be written as:(14)$$\begin{array}{ll} {\dot{V}}_{6} & ={\dot{V}}_{5}+e_{6}{\dot{e}}_{6}=-k_{5}e_{5}^{2}+e_{6}\left({\dot{e}}_{6}-e_{5}\right)\\ & =-k_{5}e_{5}^{2}+e_{6}\left({\dot{x}}_{6}-{\ddot{x}}_{5d}-k_{5}{\dot{e}}_{5}-e_{5}\right)\\ & =-k_{5}e_{5}^{2}+e_{6}\left(\frac{\cos x_{11}\cos x_{9}U_{1}}{m}-g-{\ddot{x}}_{5d}-k_{5}{\dot{e}}_{5}-e_{5}\right)\\ & <-k_{5}e_{5}^{2}-k_{6}e_{6}^{2}<0 \end{array}$$

So, we obtained this formula where $U_{1}=\frac{\left({\ddot{x}}_{5d}+g-\left(k_{5}+k_{6}\right)e_{6}+\left(1-k_{5}^{2}\right)e_{5}\right)m}{\cos x_{11}\cos x_{9}}$, and the same is true for the other control channels, all $k_{*}>0$ are parameters of control system.
(15)$$\begin{array}{l} U_{2}=\left({\ddot{x}}_{11d}-\left(k_{11}+k_{12}\right)e_{12}+\left(1-k_{11}^{2}\right)e_{11}\right)I_{x}-x_{8}x_{10}\left(I_{y}-I_{z}\right)-J_{r}x_{10}\Omega \\ U_{3}=\left({\ddot{x}}_{9d}-\left(k_{9}+k_{10}\right)e_{10}+\left(1-k_{9}^{2}\right)e_{9}\right)I_{y}-x_{8}x_{12}\left(I_{z}-I_{x}\right)-J_{r}x_{12}\Omega \\ U_{4}=\left({\ddot{x}}_{7d}-\left(k_{7}+k_{8}\right)e_{8}+\left(1-k_{7}^{2}\right)e_{7}\right)I_{z}-x_{10}x_{12}\left(I_{x}-I_{y}\right)\\ U_{5}=\left({\ddot{x}}_{1d}-\left(k_{1}+k_{2}\right)e_{2}+\left(1-k_{1}^{2}\right)e_{1}\right)m/U_{1}\\ U_{6}=\left({\ddot{x}}_{3d}-\left(k_{3}+k_{4}\right)e_{4}+\left(1-k_{3}^{2}\right)e_{3}\right)m/U_{1} \end{array}$$

Substituting Equation (15) into Equation (10), the new mathematical system model of the quadrotor aircraft is:(16)$$\left\{ \begin{array}{l} {\dot{x}}_{1}=x_{2}\\ {\dot{x}}_{2}=-\left(k_{1}+k_{2}\right)x_{2}-\left(k_{1}k_{2}+1\right)x_{1}+{\ddot{x}}_{1d}+\left(k_{1}+k_{2}\right){\dot{x}}_{1d}+\left(k_{1}k_{2}+1\right)x_{1d}\\ {\dot{x}}_{3}=x_{4}\\ {\dot{x}}_{4}=-\left(k_{3}+k_{4}\right)x_{4}-\left(k_{3}k_{4}+1\right)x_{3}+{\ddot{x}}_{3d}+\left(k_{3}+k_{4}\right){\dot{x}}_{3d}+\left(k_{3}k_{4}+1\right)x_{3d}\\ {\dot{x}}_{5}=x_{6}\\ {\dot{x}}_{6}=-\left(k_{5}+k_{6}\right)x_{6}-\left(k_{5}k_{6}+1\right)x_{5}+{\ddot{x}}_{5d}+\left(k_{5}+k_{6}\right){\dot{x}}_{5d}+\left(k_{5}k_{6}+1\right)x_{5d}\\ {\dot{x}}_{7}=x_{8}\\ {\dot{x}}_{8}=-\left(k_{7}+k_{8}\right)x_{8}-\left(k_{7}k_{8}+1\right)x_{7}+{\ddot{x}}_{7d}+\left(k_{7}+k_{8}\right){\dot{x}}_{7d}+\left(k_{7}k_{8}+1\right)x_{7d}\\ {\dot{x}}_{9}=x_{10}\\ {\dot{x}}_{10}=-\left(k_{9}+k_{10}\right)x_{10}-\left(k_{9}k_{10}+1\right)x_{9}+{\ddot{x}}_{9d}+\left(k_{9}+k_{10}\right){\dot{x}}_{9d}+\left(k_{9}k_{10}+1\right)x_{9d}\\ {\dot{x}}_{11}=x_{12}\\ {\dot{x}}_{12}=-\left(k_{11}+k_{12}\right)x_{12}-\left(k_{11}k_{12}+1\right)x_{11}+{\ddot{x}}_{11d}+\left(k_{11}+k_{12}\right){\dot{x}}_{11d}+\left(k_{11}k_{12}+1\right)x_{11d} \end{array}\right.$$

The system model of the quadrotor aircraft is simplified as:(17)$$\left\{ \begin{array}{l} \dot{x}=Ax+Bu\\ y=Cx \end{array}\right.$$
where $x={\left[\begin{array}{cccccccccccc} x & \dot{x} & y & \dot{y} & z & \dot{z} & \psi & \dot{\psi } & \theta & \dot{\theta } & \varphi & \dot{\varphi } \end{array}\right]}^{T}$, $y={\left[\begin{array}{cccccc} x & y & z & \psi & \theta & \varphi \end{array}\right]}^{T}$, A is the system matrix, B is the input matrix, and C is the output matrix.

## 3. Controller Design for Quadrotor UAV

The designed iterative learning control algorithm in Figure 2 is
(18)$$u_{k+1}=u_{k}+\varsigma e_{k}+\gamma {\dot{e}}_{k}+\eta {\ddot{e}}_{k}$$
where $e={\left[\begin{array}{cccccc} e_{x} & e_{y} & e_{z} & e_{\psi } & e_{\theta } & e_{\varphi } \end{array}\right]}^{T}$, $\dot{e}={\left[\begin{array}{cccccc} {\dot{e}}_{x} & {\dot{e}}_{y} & {\dot{e}}_{z} & {\dot{e}}_{\psi } & {\dot{e}}_{\theta } & {\dot{e}}_{\varphi } \end{array}\right]}^{T}$, $\ddot{e}={\left[\begin{array}{cccccc} {\ddot{e}}_{x} & {\ddot{e}}_{y} & {\ddot{e}}_{z} & {\ddot{e}}_{\psi } & {\ddot{e}}_{\theta } & {\ddot{e}}_{\varphi } \end{array}\right]}^{T}$, and $u={\left[\begin{array}{cccccc} u_{x} & u_{y} & u_{z} & u_{\psi } & u_{\theta } & u_{\varphi } \end{array}\right]}^{T}$. The three iterative learning gain matrices are expressed as $\varsigma =\varsigma_{0}+\varsigma_{m}$, $\gamma =\gamma_{0}+\gamma_{m}$, $\eta =\eta_{0}+\eta_{m}$. $\varsigma_{0}$, $\gamma_{0}$, and $\eta_{0}$ are the initial given values, and the fuzzy controller is used to adjust $\varsigma_{m}$, $\gamma_{m}$, and $\eta_{m}$.

The fuzzy controller in this paper has three parameters $\varsigma_{m}$, $\gamma_{m}$, and $\eta_{m}$ as the output. $e\left(t\right)=\left[\begin{array}{cccccc} e_{x}\left(t\right) & e_{y}\left(t\right) & e_{z}\left(t\right) & e_{\psi }\left(t\right) & e_{\theta }\left(t\right) & e_{\varphi }\left(t\right) \end{array}\right]$, $e_{x}\left(t\right)=x_{d}\left(t\right)-x_{k}\left(t\right)$, $e_{y}\left(t\right)=y_{d}\left(t\right)-y_{k}\left(t\right)$, $e_{z}\left(t\right)=z_{d}\left(t\right)-z_{k}\left(t\right)$, $e_{\psi }\left(t\right)=\psi_{d}\left(t\right)-\psi_{k}\left(t\right)$, $e_{\theta }\left(t\right)=\theta_{d}\left(t\right)-\theta_{k}\left(t\right)$, and $e_{\varphi }\left(t\right)=\varphi_{d}\left(t\right)-\varphi_{k}\left(t\right)$. $x_{d}\left(t\right)$, $y_{d}\left(t\right)$, $z_{d}\left(t\right)$ and $x_{k}\left(t\right)$, $y_{k}\left(t\right)$, $z_{k}\left(t\right)$ represent the expected position coordinates and the actual position coordinates of the quadrotor aircraft of the k-th iteration, respectively. $\psi_{d}\left(t\right)$, $\theta_{d}\left(t\right)$, $\varphi_{d}\left(t\right)$ and $\psi_{k}\left(t\right)$, $\theta_{k}\left(t\right)$, $\varphi_{k}\left(t\right)$ represent the desired attitude angle and the actual attitude angle of the quadrotor aircraft in the k-th iteration, respectively. The fuzzy rules table is shown in Table 2. In these figures, PB is positive big, PM is positive middle, PS is positive small, NB is negative big, and ZO is zero. The rules of the controllers are expressed in Table 2 with all the possible combinations. Based on the control experience of the four-rotor aircraft, the fuzzy rules were established according to the output error of the system and the adjustment of the parameters. The membership function of the fuzzy controller is shown in Figure 3.

## 4. Convergence Analysis

In this section, the convergence condition of the controllers for the quadrotor aircraft system is given and proved.

**Theorem** **1.**
*The quadrotor aircraft system in Equation (17) meets the conditions:*
(19)$$(1)\;\left\|I-CB\left(\eta_{0}+\eta_{m}\right)\right\|\leq \bar{\rho }<1$$
(20)$$(2)\;x_{k}\left(0\right)=x^{0},\;e_{k}\left(0\right)=0,\;{\dot{e}}_{k}\left(0\right)=0,\;k=0,1,2,\cdots$$

*Moreover, an approximation to the value of*
$y_{k}\left(t\right)\to y_{d}\left(t\right)$
*, obtained long before exact termination should occur, is often sufficient. Therefore, the following iteration termination criterion is chosen:*
(21)$$\left\|y_{d}\left(t\right)-y_{k}\left(t\right)\right\|\leq \varepsilon$$
*where*
ε>0
*is a strict accuracy bound. After*
k
*iterations, it is possible to obtain an approximation*
$y_{k}\left(t\right)$
*to*
$y_{d}\left(t\right)$
*from the iteration procedure. Under the action of the proposed algorithm in Equation (18), when*
k→∞
*, we obtained the conclusion of this theorem*
$y_{k}\left(t\right)\to y_{d}\left(t\right)$
*,*
t∈[0,T]
*.*


**Proof.** The error variables are ${\dot{e}}_{k}\left(t\right)={\dot{y}}_{d}\left(t\right)-{\dot{y}}_{k}\left(t\right)$
(22)$$\begin{array}{ll} {\dot{e}}_{k+1}\left(t\right) & ={\dot{e}}_{k}\left(t\right)-\int_{0}^{t}C\Phi \left(t,\tau \right)B\left(u_{k+1}\left(\tau \right)-u_{k}\left(\tau \right)\right)d\tau \\ & ={\dot{e}}_{k}\left(t\right)-\int_{0}^{t}C\Phi \left(t,\tau \right)B\left(\left(\varsigma_{0}+\varsigma_{m}\right)\int_{0}^{\tau }{\dot{e}}_{k}\left(\sigma \right)d\sigma +\left(\gamma_{0}+\gamma_{m}\right){\dot{e}}_{k}\left(\tau \right)+\left(\eta_{0}+\eta_{m}\right){\ddot{e}}_{k}\left(\tau \right)\right)d\tau \end{array}$$By integration by parts, the formula can be obtained:(23)$$\begin{array}{ll} \int_{0}^{t}G\left(t,\tau \right){\ddot{e}}_{k}\left(\tau \right)d\tau & =G\left(t,\tau \right){\dot{e}}_{k}\left(\tau \right)|{}_{0}^{t}-\int_{0}^{t}\frac{\partial G\left(t,\tau \right)}{\partial t}{\dot{e}}_{k}\left(\tau \right)d\tau \\ & =C\left(t\right)B\left(t\right)\left(\eta_{0}+\eta_{m}\right){\dot{e}}_{k}\left(t\right)-\int_{0}^{t}\frac{\partial G\left(t,\tau \right)}{\partial t}{\dot{e}}_{k}\left(\tau \right)d\tau \end{array}$$
where $G\left(t,\tau \right)=C\Phi \left(t,\tau \right)B\left(\eta_{0}+\eta_{m}\right)$.Substitute Equation (23) into Equation (22)
(24)$$\begin{array}{c} {\dot{e}}_{k+1}\left(t\right)=\left[I-C\left(t\right)B\left(t\right)\left(\eta_{0}+\eta_{m}\right)\right]{\dot{e}}_{k}\left(t\right)+\int_{0}^{t}\frac{\partial G\left(t,\tau \right)}{\partial t}{\dot{e}}_{k}\left(\tau \right)d\tau -\\ \int_{0}^{t}C\Phi \left(t,\tau \right)B\left(\tau \right)\left(\gamma_{0}+\gamma_{m}\right){\dot{e}}_{k}\left(\tau \right)d\tau \\ -\int_{0}^{t}\int_{0}^{\tau }C\Phi \left(t,\tau \right)B\left(\tau \right)\left(\varsigma_{0}+\varsigma_{m}\right){\dot{e}}_{k}\left(\sigma \right)d\sigma d\tau \end{array}$$Then, the norm of both sides of Equation (24) can be obtained:(25)$$\begin{array}{ll} \left\|{\dot{e}}_{k+1}\left(t\right)\right\| & \leq \left\|I-C\left(t\right)B\left(t\right)\left(\eta_{0}+\eta_{m}\right)\right\|\left\|{\dot{e}}_{k}\left(t\right)\right\|+\int_{0}^{t}\left\|\frac{\partial G\left(t,\tau \right)}{\partial t}\right\|\left\|{\dot{e}}_{k}\left(\tau \right)\right\|d\tau \\ & +\int_{0}^{t}\left\|C\Phi \left(t,\tau \right)B\left(\tau \right)\left(\gamma_{0}+\gamma_{m}\right)\right\|\left\|{\dot{e}}_{k}\left(\tau \right)\right\|d\tau \\ & +\int_{0}^{t}\int_{0}^{\tau }\left\|C\Phi \left(t,\tau \right)B\left(\tau \right)\left(\varsigma_{0}+\varsigma_{m}\right)\right\|\left\|{\dot{e}}_{k}\left(\sigma \right)\right\|d\sigma d\tau \\ & \leq \left\|I-C\left(t\right)B\left(t\right)\left(\eta_{0}+\eta_{m}\right)\right\|\left\|{\dot{e}}_{k}\left(t\right)\right\|+\int_{0}^{t}b_{1}\left\|{\dot{e}}_{k}\left(\tau \right)\right\|d\tau +\int_{0}^{t}\int_{0}^{\tau }b_{2}\left\|{\dot{e}}_{k}\left(\sigma \right)\right\|d\sigma d\tau \end{array}$$
where $b_{1}=\max\left\{\underset{t,\tau \in \left[0,T\right]}{\sup}\left\|\frac{\partial G\left(t,\tau \right)}{\partial t}\right\|,\underset{t,\tau \in \left[0,T\right]}{\sup}\left\|C\Phi \left(T,\tau \right)B\left(\tau \right)\left(\gamma_{0}+\gamma_{m}\right)\right\|\right\}$ and $b_{2}=\underset{t,\tau \in \left[0,T\right]}{\sup}\left\|C\Phi \left(t,\tau \right)B\left(\tau \right)\left(\varsigma_{0}+\varsigma_{m}\right)\right\|$. Multiplying both sides of Equation (14) by $e^{-\lambda t}$ to compute the λ-norm, we obtain:(26)$${\left\|\dot{e}{}_{k+1}\right\|}_{\lambda }\leq \tilde{\rho }{\left\|\dot{e}{}_{k}\right\|}_{\lambda }$$
where $\tilde{\rho }=\bar{\rho }+b_{1}\left(1-e^{-\lambda t}\right)/\lambda +b_{2}{\left(1-e^{-\lambda t}\right)}^{2}/\lambda^{2}$. We found a sufficiently large positive number λ, so we obtained $\tilde{\rho }<1$. Therefore, we could reasonably choose a group control parameter in order to reach the conclusion of this theorem $\underset{k\to \infty }{\lim}{\left\|\dot{e}{}_{k}\right\|}_{\lambda }=0$. □

## 5. Gazebo Environment Simulation

To demonstrate the tracking performance and robustness of the proposed ILC law, the overall system was tested using Gazebo simulations. The modeling of the Gazebo simulation does not depend on the mathematical model of the quadrotor itself or any special graphics package and can simulate various dynamic relationships between spatial objects in virtual space, which has the advantages of other simulation software. The physical parameters of the whole quadrotor system are shown in Table 1. The parameters of the fuzzy control laws are listed in Table 2. The simulation time is 45 s. The desired helical trajectory is $P_{d}=\left[\begin{array}{ccc} t/2\cos\left(t/2\right) & t/2\sin\left(t/2\right) & t/10 \end{array}\right]$. The external aerodynamic interference during the quadrotor flight is: $d_{f}=\left[\begin{array}{ccc} 0.1\sin\left(0.1\pi t\right) & 0.15\cos\left(0.1\pi t\right) & 0.12\cos\left(0.1\pi t\right) \end{array}\right]$.

Figure 4 presents the model of the quadrotor aircraft in the Gazebo simulation environment. Figure 5 and Figure 6 show the three-dimensional (3D) trajectory tracking the result of the quadrotor. We can see almost asymptotic convergence toward the actual tracking trajectory after 10 iterations. Simulation results for each direction of the reference trajectories and initial conditions showed better tracking results. The fuzzy PID-ILC demonstrated remarkable performance in terms of control and stability of the system compared with the conventional PID-ILC algorithm. The maximum tracking errors in the $x,y,z,\varphi ,\theta_{}\;\mathrm{and}\;\psi$ directions from iteration to iteration are depicted in Figure 7. Figure 8 shows the tracking errors in the last iteration controlled by both fuzzy PID-ILC and traditional PID-ILC. The fuzzy PID-ILC performed much better than the PID-ILC in terms of the convergence speed and tracking error reductions. In the presence of wind disturbances, there are smaller errors for the motions in all three directions controlled by the fuzzy PID-ILC. These results show the importance of fuzzy control. Therefore, the fuzzy PID-ILC algorithm has indicated its capability to solve the trajectory-tracking control problem experienced by quadrotor UAVs.

## 6. Conclusions

The novel fuzzy PID-ILC algorithm was successfully applied to the trajectory tracking of a quadrotor UAV. A simple fuzzy law to tune the PID-ILC gains was developed. The PID-ILC algorithm adjusts and produces a group of the optimal input compensation for each iteration so that the overall error is reduced and converges to a minimized tracking error. By comparing the results of the Gazebo simulation, fuzzy PID-ILC demonstrated its remarkable capability to not only maintain the stability of the system and reduce the shaking and concussion of the system but also to achieve perfect tracking of the trajectory. Future research directions will include applications of the fuzzy iterative learning algorithm for the selection of the controller parameters.

## Figures and Tables

**Figure 1 sensors-19-00024-f001:**
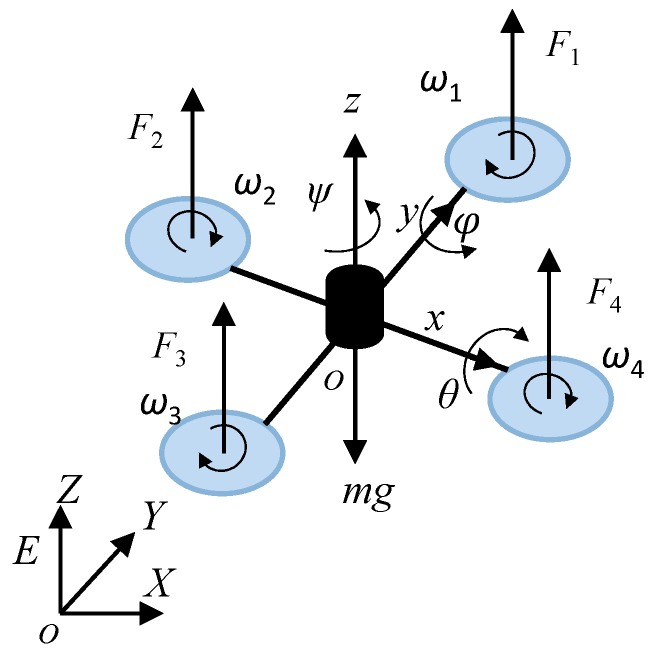
Quadrotor structure.

**Figure 2 sensors-19-00024-f002:**
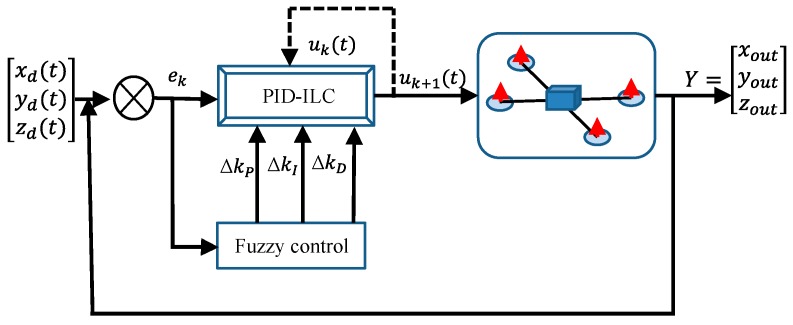
System architecture of the fuzzy PID-ILC for the quadrotor.

**Figure 3 sensors-19-00024-f003:**
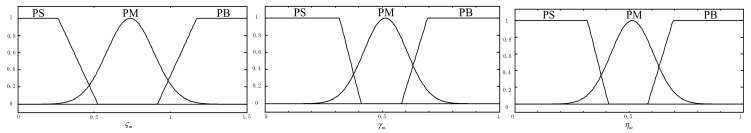
Fuzzy membership functions.

**Figure 4 sensors-19-00024-f004:**
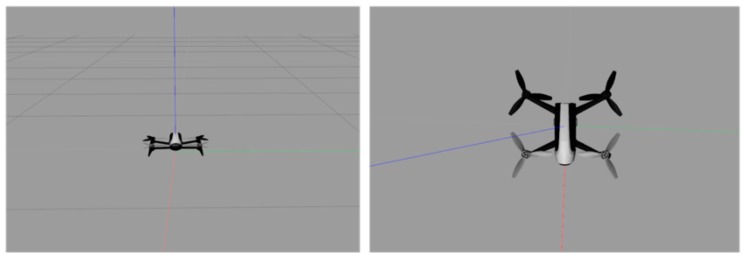
The model of quadrotor aircraft in the Gazebo simulation environment.

**Figure 5 sensors-19-00024-f005:**
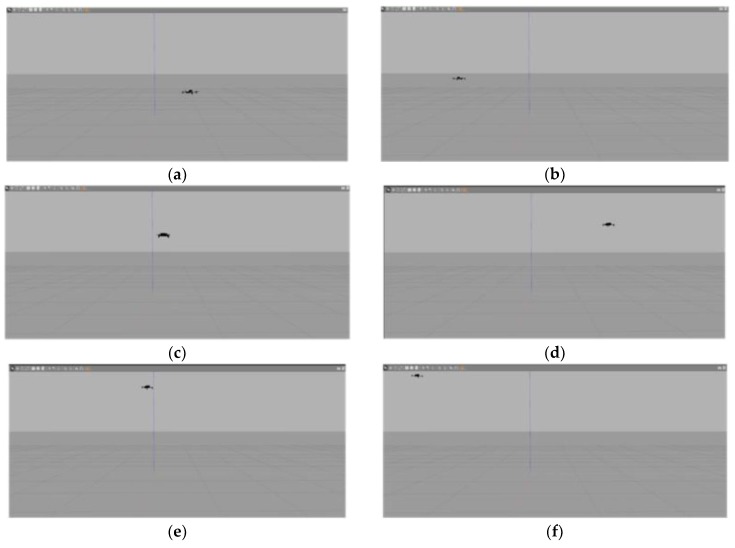
The flying process of the quadrotor aircraft in the Gazebo simulation environment.

**Figure 6 sensors-19-00024-f006:**
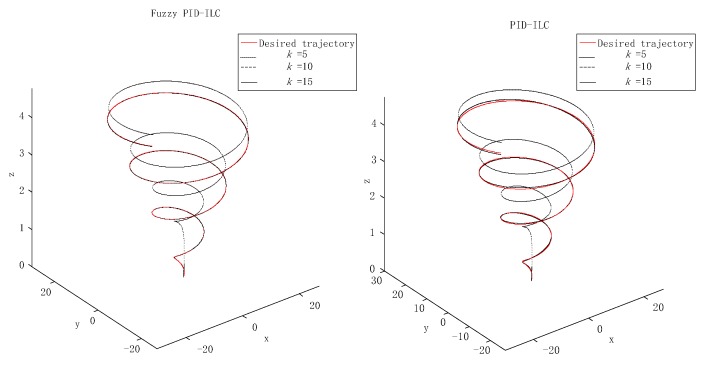
Trajectory of the quadrotor flight.

**Figure 7 sensors-19-00024-f007:**
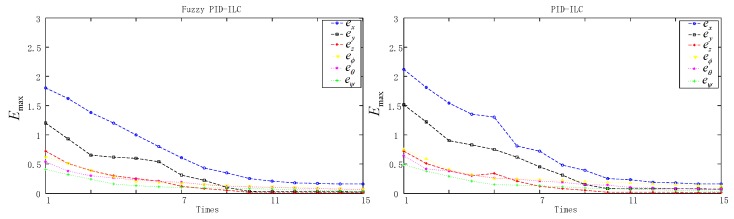
Maximum absolute values of the tracking error.

**Figure 8 sensors-19-00024-f008:**
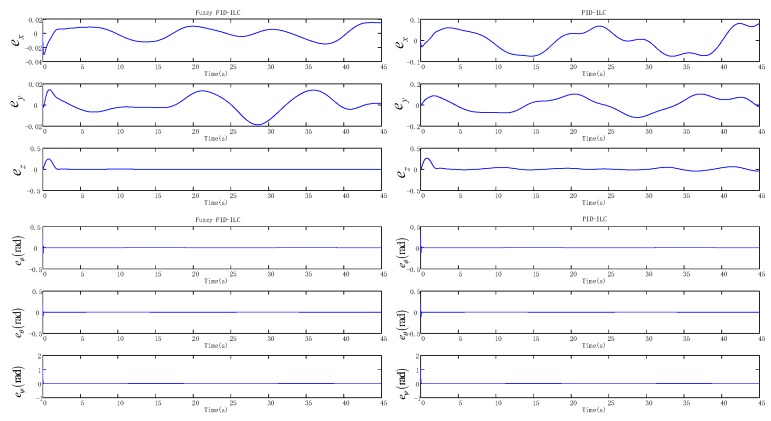
The changing curves of the tracking errors in the final iteration.

**Table 1 sensors-19-00024-t001:** Parameters of the quadrotor unmanned aerial vehicle (UAV).

Parameter	Description	Value	Unit
*m*	Total quadrotor mass	1	kg
*l*	Quadrotor radius length	0.25	m
*I_x_*	Moment of inertia about *X*-axis	4 × 10^−3^	Kg·m^2^
*I_y_*	Moment of inertia about *Y*-axis	4 × 10^−3^	kg·m^2^
*I_z_*	Moment of inertia about *Z*-axis	8 × 10^−3^	kg·m^2^
*ω* _max_	Maximum rotor speed	200	rad/s
*g*	Gravitational acceleration	9.81	ms^2^

**Table 2 sensors-19-00024-t002:** Fuzzy rules.

$\varsigma_{m}$ $/\gamma_{m}$ $/\eta_{m}$		*e*		
		NB	ZO	PB
$e_{\theta }$	**NB**	PB/PS/PM	PB/PS/PS	PB/PS/PS
	**ZO**	PM/PM/PB	PS/PB/PM	PM/PM/PB
	**PB**	PB/PS/PS	PB/PS/PS	PB/PS/PM

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
