# Peer review of "Novel Fuzzy PID-Type Iterative Learning Control for Quadrotor UAV"

_sensors, 2018, doi:10.3390/s19010024_

Reviewer 2 Report

The authors should explain clearly what is the "problem of chattering caused by the external disturbance". It is supposed that chattering is introduced by discontinuous controllers when trying to compensate disturbances. 

Authors should perform simulations with other control approaches in order to show how good is the new scheme. 

Authors do not explain details of the experimental system. Usually, the Parrot Bebop 2 does not provide a direct access to the signal of position and orientation. Besides, how the control action is commanded? I suggest to remove the Section "Experiments on the quadrotor" since is doubtful. Usually position information is very noisy and figure 10 looks like a numerical simulation.

Literature review is very limited. There are many PID controllers for quadrotors recently published that authors should include in the literature review.

Author Response

Round  2

Reviewer 1 Report

Comments and recommendations are included in the revised manuscript.

Author Response

Thank you for your comments and recommendations. 

Our modifications  are included in the revised manuscript.

Reviewer 2 Report

Paper has improved but some references should be added

Moreno-Valenzuela, J., Pérez-Alcocer, R., Guerrero-Medina, M., & Dzul, A. (2018). Nonlinear PID-Type Controller for Quadrotor Trajectory Tracking. IEEE/ASME Transactions on Mechatronics23(5), 2436-2447.

Qiao, J., Liu, Z., & Zhang, Y. (2018, February). Modeling and GS-PID Control of the Quad-Rotor UAV. In Proceedings of the 2018 10th International Conference on Computer and Automation Engineering (pp. 221-226). ACM.

Cherrat, N., Boubertakh, H., & Arioui, H. (2018, February). Adaptive fuzzy PID control for a quadrotor stabilisation. In IOP Conference Series: Materials Science and Engineering (Vol. 312, No. 1, p. 012002). IOP Publishing.

Author Response

Thank you for your comments.   As Reviewer suggested that some references were added.

Moreno-Valenzuela, J., Pérez-Alcocer, R., Guerrero-Medina, M., & Dzul, A. (2018). Nonlinear PID-Type Controller for Quadrotor Trajectory Tracking. IEEE/ASME Transactions on Mechatronics23(5), 2436-2447.

Qiao, J., Liu, Z., & Zhang, Y. (2018, February). Modeling and GS-PID Control of the Quad-Rotor UAV. In Proceedings of the 2018 10th International Conference on Computer and Automation Engineering (pp. 221-226). ACM.

Cherrat, N., Boubertakh, H., & Arioui, H. (2018, February). Adaptive fuzzy PID control for a quadrotor stabilisation. In IOP Conference Series: Materials Science and Engineering (Vol. 312, No. 1, p. 012002). IOP Publishing.